# Protocol of BEYOND trial: Clinical BEnefit of sodium-glucose cotransporter-2 (SGLT-2) inhibitors in rhYthm cONtrol of atrial fibrillation in patients with diabetes mellitus

**Kyuhyun Lee**[1⊙], **Soo Kyoung Lee**[1⊙], **Juyeon Lee**[1⊙], **Bo Kyung Jeon**[2⊙], **Tae-Hoon Kim**[3], **Hee Tae Yu**[3], **Jung Myung Lee**[4], **Jin-Kyu Park**[5], **Yong-Soo Baek**[6], **Dong Hyeok Kim**[2], **Jaemin Shim**[7], **Boyoung Joung**[3], **Moon-Hyoung Lee**[3], **Hui-Nam Pak**[3], **Junbeom Park**[2]*

1 College of Medicine, Ewha Womans University, Seoul, Republic of Korea, 2 Department of Cardiology, Ewha Womans University Medical Center, Seoul, Republic of Korea, 3 Yonsei University Health System, Seoul, Republic of Korea, 4 Kyung Hee University, Medical Center, Seoul, Republic of Korea, 5 Department of Cardiology, Hanyang University Seoul Hospital, Seoul, Republic of Korea, 6 Inha University College of Medicine and Inha University Hospital, Incheon, Republic of Korea, 7 Korea University College of Medicine, Anam Hospital, Seoul, Republic of Korea

⊙ These authors contributed equally to this work.

* parkjb@ewha.ac.kr

**Data Availability Statement:** No datasets were generated or analysed during the current study. All

## Abstract

SGLT-2 inhibitor, traditionally used for glycemic control, has several beneficial effects that can help manage heart failure (HF). SGLT-2 inhibitors reduce the risk of cardiovascular mortality in patients with HF. As atrial fibrillation (AF) is closely associated with HF and diabetes mellitus (DM) is a risk factor for AF, we assume that SGLT-2 inhibitors will also show therapeutic benefits regarding AF, especially for rhythm control. This trial has a multicenter, prospective, open, blinded endpoint design. It is a 1:1 randomized and controlled study. A total of 716 patients who are newly diagnosed of AF and DM within 1 year will be enrolled from 7 tertiary medical centers. The trial is designed to compare the effects of SGLT-2 inhibitors and other oral hypoglycemic agents on atrial rhythm control in patients with AF and DM. The primary outcome is the recurrence of AF within a year (including post-antiarrhythmic drugs (AAD) or ablation). The secondary outcomes are the ablation rate within a year, change in AF burden, size of the left atrium, NT-proBNP, the AF symptom score, and the quality of life. This trial will prospectively evaluate the effect and safety of SGLT-2 inhibitors on AF rhythm control in patients with DM. It will provide an invaluable dataset on rhythm control in AF with DM for future studies and offer novel information to assist in clinical decisions. (BEYOND trial, ClinicalTrials.gov number: NCT05029115. https://clinicaltrials.gov/ct2/show/NCT05029115).

## Introduction

Atrial fibrillation (AF) is a common cause of cardiac arrhythmia, and its prevalence increases with age [1,2]. In the United States alone, the prevalence of AF is expected to increase to 12.1

relevant data from this study will be made available upon study completion.

**Funding:** This trial is supported by Basic Science Research Program through the National Research Foundation of Korea (NRF) funded by the Ministry of Science, ICT & Future Planning (NRF-2017R1E1A1A01078382), and by the Korea Medical Device Development Fund grant funded by the Republic of Korea government (the Ministry of Science and ICT, the Ministry of Trade, Industry and Energy, the Ministry of Health & Welfare, the Ministry of Food and Drug Safety) (Project Number: 9991006899). The funders had no role in study design, data collection and analysis, decision to publish, or preparation of the manuscript.

**Competing interests:** The authors have declared that no competing interests exist.

million in 2030 from 5.2 million in 2010 [3]. In addition, owing to the growth in the global prevalence of diabetes mellitus (DM) [4], it has become necessary to manage both AF and DM simultaneously. The Action in Diabetes and Vascular Disease: preterAx and diamicroN-MR Controlled Evaluation (ADVANCE) study showed that AF is relatively common in DM and is associated with a considerably increased risk of cardiovascular events and death in patients with DM [5]. According to the Outcomes Registry for Better Informed Treatment of Atrial Fibrillation (ORBIT-AF) registry, DM is not only a risk factor for AF, but also associated with worse AF symptoms and a lower quality of life [6].

AF and congestive heart failure (CHF) are commonly encountered disease entities that share common risk factors such as hypertension, DM, ischemic heart disease, and valvular heart disease [7,8]. AF and CHF are so intertwined that many questions related to their pathophysiologic relationship are still unknown [9]. It has been suggested that heart failure (HF) increases the risk of AF by elevating atrial filling pressures [10,11], inducing alterations in intracellular calcium [12,13], and disturbing the neuro-endocrinological balance [7]. The Dapagliflozin and Prevention of Adverse Outcomes in Heart Failure (DAPA-HF) trial and The Empagliflozin Outcome Trial in Patients with Chronic Heart Failure and Reduced Ejection Fraction (EMPEROR-Reduced) trial revealed that two sodium–glucose cotransporter 2 (SGLT-2) inhibitors, dapagliflozin and empagliflozin, reduce the risk of cardiovascular death or HF exacerbation, regardless of the presence or absence of DM [14,15]. Besides glucose-lowering, SGLT-2 inhibitors appear to have several other beneficial effects as well.

Recent meta-analyses and retrospective studies have reported that the use of SGLT-2 inhibitors in type 2 DM patients with or without HF, as well as in type 2 DM patients with HF, reduces the incidence rate of AF, atrial flutter, and other cardiovascular events [16–19]. However, few randomized controlled studies have reported the relationship between SGLT-2 and AF, and therefore, reports of detailed prognostic characteristics such as change of AF burden and rhythm control other than the incidence of AF are rare. This trial aimed to determine whether SGLT-2 inhibitors have an advantageous influence on rhythm control of AF and, hence, reduce the adverse outcomes of AF through prospective, randomized, and controlled methods.

## Materials and methods

### Study hypothesis and primary outcome

The primary hypothesis of this study is that the SGLT-2 inhibitor is superior to other oral hypoglycemic medications for rhythm control in patients with AF and DM, diagnosed within the prior year. A step-up treatment will be carried out when the participant's AF recurs. The recurrence rate of AF will be investigated after performing stepwise rhythm control therapies including anti-arrhythmic drugs (AAD) and ablation. AF is defined as per the 2019 AHA/ACC/HRS Guideline, i.e., irregular R–R intervals (when atrioventricular conduction is present), absence of distinct repeating P waves, and irregular atrial activity [20]. It is checked by 24-h Holter ECG every 3 months and is considered significant when it lasts for > 30 s [21].

### Secondary outcomes

Secondary outcomes are listed in Table 1. AF-free survival will be compared between the two groups as the first of the secondary outcome, analyzed using the Kaplan–Meier method, and documented on curve. AF burden at 3-month follow-up visit, 12-month follow-up visit in those who did not step up and used AAD only for rhythm control, immediately before trying ablation, and at 12-month follow-up visit in those who stepped up and underwent ablation in both the SGLT-2 inhibitor group and the control group will be compared. AF burden is

**Table 1. Secondary outcomes.**

- Rhythm control
  - AF-free survival
  - AF burden at
    - 3-month F/U visit, 12-month F/U visit in those who did not step up to session 2
    - Immediately before trying ablation, 12-month F/U visit in those who stepped up to session 2
  - Percentage of patients undergoing ablation within a year
  - Sinus rhythm maintenance*
- Diameter of LA
- NT-proBNP
- Symptom score (EHRA score)
- Quality of life (EQ-5D, SF-12)

*Defined as the absence of clinically relevant arrhythmia (e.g., AF, AFL, sustained VT, and VF).

AF, atrial fibrillation; LA, left atrium; F/U, follow up; NT-proBNP, N-terminal pro-B-type natriuretic peptide; EHRA, European Heart Rhythm Association; EQ-5D, EuroQol-5 Dimension; SF-12, Short Form-12 Health Survey Questionnaire; AFL, atrial flutter; VT, ventricular tachycardia; VF, ventricular fibrillation.

defined as the atrial fibrillation time percentage documented on 24-h Holter ECG [22]. Ablation rate, which is the percentage of patients undergoing ablation, within a year will also be compared between the two groups to identify the efficacy of symptom and rhythm control using SGLT-2 inhibitors. Sinus rhythm is considered stable when either a standard ECG or a 24-h Holter ECG showed no episode of clinically relevant arrhythmia, including AF, at the time of check-up [23]. The proportion of participants with stable sinus rhythm in all participants, regardless of with or without ablation, is compared between the two groups at baseline, 6-, and 12-month follow-up visits.

As dilatation of the left atrium (LA) is associated with AF in both pathophysiologic and prognostic perspectives, we will use the LA size as a secondary outcome. Left atrial remodeling induces electrical modification, which is associated with AF. LA structural remodeling is characterized by atrial dilatation [24]. An increase in the LA size at baseline is associated with progressive LA enlargement and AF recurrence. Hence, the LA size can be considered a prognostic factor for spontaneous conversion of AF [25]. As an indicator of LA dilatation, the left atrial diameter (anterior–posterior) will be measured by transthoracic echocardiography (TTE) at baseline and 12-month follow-up visit [26]. N-terminal pro-B-type natriuretic peptide (NT-proBNP) is often elevated in AF and is higher in severe AF [27,28]. As it is a significant secondary outcome, we will measure the level of NT-proBNP in blood and compare their values in the two groups at baseline and 12-month follow-up visit.

The symptom score is compared by using the EHRA score. It is measured at baseline, 3-, 6-, 9-, and 12-month of follow-up visits. The ratio of asymptomatic participants will be compared between the two groups. To compare the quality of life between the two groups, EuroQoL five-dimensional instrument (EQ-5D, www.euroqol.org) and SF-12 (a shortened version of SF-36, www.sf-36.org) Health Survey will be used, and assessments will be conducted between the data on baseline and 12-month follow-up visit.

## Adverse events

Adverse events are assessed and judged twice a year by an independent committee. More than half of the votes should be matched to finalize the decision. Adverse events are listed in Table 2. Any unexpected medical events or laboratory findings are also defined as adverse events. Atrial arrhythmia alone is not classified as an adverse event. Instead, it is considered as the primary and/or secondary outcome.

**Table 2. Adverse event and safety outcome.**

<u>Adverse event</u>
- Death caused by
  - The cardiovascular event
  - Other causes related to therapy and underlying disease(s)
- Life-threatening events related to the cardiovascular cause, therapy, or underlying disease(s) determined based on clinical decision
- Any unexpected medical events or laboratory findings

<u>Safety outcome</u>
- Genitourinary infection*
- Renal dysfunction**
- Symptoms of volume depletion
- Diabetic ketoacidosis
- Hypoglycemia
- Fracture
- Lower limb amputation

\* Symptomatic genitourinary infection as at least one event.

\*\* Deterioration of renal function, change in the Chronic Kidney Disease stage (KDIGO 2021) [29].

## Study design and participants

This is a multicenter study with a prospective, open, blinded endpoint design (Fig 1). It is a 1:1 randomized and controlled study that compares the rhythm control effect of SGLT-2 inhibitor therapy to other oral hypoglycemic agents in patients with AF and DM (Fig 2).

Patients diagnosed with AF within a year, who have never undergone any ablation treatment, newly diagnosed with DM, and with ages between 18 and 80 years will be selected as the study participants. They will be enrolled in 7 tertiary medical centers from the Division of Cardiology and the Division of Endocrinology. As patients diagnosed with AF also tend to check for DM to calculate the CHAD2-VASc score, those who have been diagnosed as AF in the prior year and newly diagnosed with DM in the follow-up can be recruited for the trial in the Division of Cardiology. To identify patients for enrollment in the Division of Endocrinology, every patient newly diagnosed with DM during the enrollment period will be asked for 12-lead ECG and 24-h Holter ECG; those with AF > 10 s in 12-lead ECG or >30 s in 24-h Holter ECG can be recruited for the trial. All participants must meet the required inclusion (Table 3) and exclusion criteria (Table 4) at the point of randomization. The participants should otherwise be healthy and use only oral hypoglycemic agents, not subcutaneous agents, for glycemic control.

The rhythm control strategy for AF will be followed, independent of the type of hypoglycemic agents received. This strategy includes two sessions (Fig 2). In session 1, an AAD will be assigned to every participant and taken for at least 3 months. If patients have persistent AF, cardioversion will be performed with AAD. During the follow-up, those with recurrent AF should step up to session 2, which involves treatment with radiofrequency ablation (RFA) or CRYO-balloon ablation (4 pulmonary vein isolation; 4PVI). For participants of session 2, the follow-up will begin with the moment RFA or cryotherapy is applied, which will be regarded as 0-month; follow-up will be conducted every 3 months for the next 12 months. The rhythm control strategy will be conducted as per the clinical guidelines of 2020 ESC [30] and 2019 AHA/ACC/HRS guidelines [20] for AF.

## Treatment: DM control (allocation)

The subjects will receive oral hypoglycemic agents for glucose control; the type of agents will be determined by randomization. Patients using SGLT-2 inhibitors to control their serum

| | STUDY PERIOD | | | | | | |
|---|---|---|---|---|---|---|---|
| | Enrolment & Allocation | | | | Post-allocation | | Close-out |
| TIMEPOINT | 2021/10 ~ 2022/12 | | 2023/01 ~ 2023/12 | | 2024/01 ~ 2025/12 | | 2026/01 ~ 2026/12 |
| *ENROLMENT:* | | | | | | | |
| Eligibility screen | X | X | X | X | | | |
| Informed consent | X | X | X | X | | | |
| Allocation | X | X | X | X | | | |
| *INTERVENTIONS:* | | | | | | | |
| Adjusted DM medication | X | X | X | X | X | X | |
| AAD (± cardioversion) | X | X | X | X | X | X | |
| Follow-up (Consider ablation if AF is recurred.) | | | X | X | X | X | |
| *ASSESSMENTS:* | | | | | | | |
| Result analysis | | | | | | | X |
| Report | | | | | | | X |

Abbreviations: DM, diabetes mellitus; AAD, Anti-arrhythmic drugs; AF, atrial fibrillation

**Fig 1. Completed SPIRIT schedule of trial.** The rough schedule of BEYOND trial is summarized in a table. If AF is recurred during the follow-up period, patient will get a catheter or CRYO-balloon ablation and have an additional one-year follow-up period after ablation.

glucose will be specified as the case group, while those using other hypoglycemic agents (e.g., metformin, meglitinides, sulfonylureas, DPP-4 inhibitors, GLP-1 receptor agonists, α-glucosidase inhibitors, and thiazolidinediones) to control their serum glucose will be classified as the control group. The number, type, and dosage of the agents will be individualized at the discretion of the physician in accordance with the 2021 ADA guidelines [31], considering factors such as the kidney function, age, and cardiovascular status.

## Follow-up and detection of AF recurrence

After their recruitment, the participants will receive baseline and evaluation studies under scrutiny, which include history taking, physical examination, blood sampling, urine sampling, 12-lead ECG, 24-h Holter ECG, TTE, EHRA, EQ-5D, and SF-12. The follow-up schedule will begin with the prescription of oral hypoglycemic agents (Table 5). During each intervention, the following steps were conducted: (1) history taking for the review of AF and DM status, adverse events, and other specific illnesses on every visit; (2) physical examination to evaluate any DM complication, AF, NYHA score, neurologic exam, systolic and diastolic blood

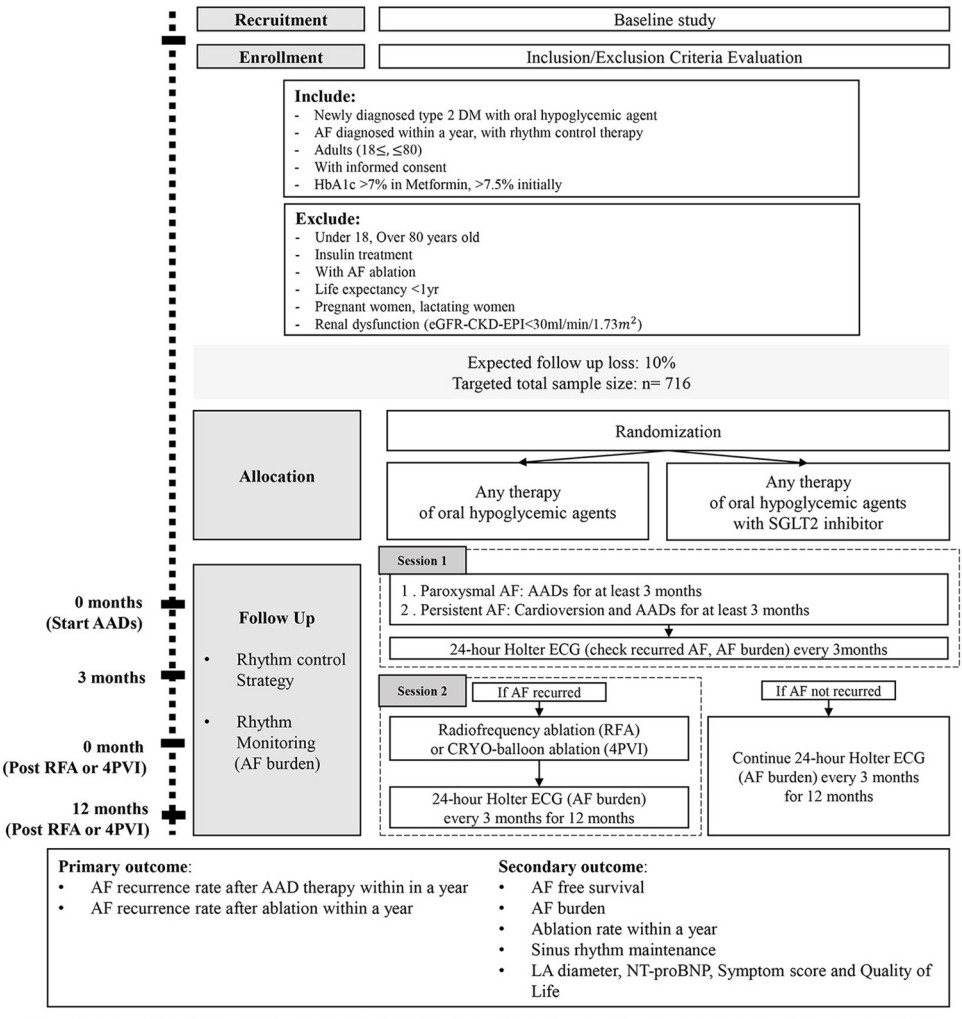

Abbreviations: DM, diabetes mellitus; AF, atrial fibrillation; EPI, epidemiology collaboration; AAD, Anti-arrhythmic drugs; ECG, electrocardiogram; RFA, radiofrequency ablation; 4PVI, 4 pulmonary vein isolation; LA, Left atrium; NT-proBNP, N-terminal pro-B-type natriuretic peptide

**Fig 2. Flow chart of the trial.** Brief flow of the BEYOND trial described as a schematic diagram. Patients who enroll this trial will be randomly administrated SGLT-2 inhibitor or other DM medication while applying stepwise rhythm control strategy for AF.

**Table 3. Inclusion criteria.**

• At least one episode of AF* that is documented during the prior year by any kind of ECG recording.
• Type 2 DM was diagnosed (HbA1c > 6.5%) and the patient was using oral hypoglycemic agents only for glycemic control.
• At least 18 years old, but not older than 80 years.
• Normal ECG parameters, measured in sinus rhythm (QRS width $\leq$ 120 ms, QTc interval < 440 ms, and PQ interval $\leq$ 210 ms in a 12-lead ECG).
• Be able and willing to give informed consent.

*AF episode must last longer than 30 s on single ECG, 10 s on 12-lead ECG or 24-h Holter ECG.

AF, atrial fibrillation; DM, diabetes mellitus; ECG, electrocardiogram.

**Table 4. Exclusion criteria.**

- Any disease that limits life expectancy to under 1 year
- Subject for another clinical trial within the past 2 months
- Under 18 years old or over 80 years
- Pregnant women
- Lactating women
- Drug abuser
- Type 2 DM treated by recombinant insulin
- Diagnosis of Type 1 DM, MODY, or secondary DM
- HbA1c $\geq$ 12% or HbA1c < 6.5% at diagnosis
- Previous treatment with any SGLT-2 inhibitor
- Renal dysfunction (eGFR-CKD-EPI < 30 mL/min/1.73 m$^2$)
- Chronic cystitis and/or recurrent genitourinary tract infections (3 or more in the last year)
- Unexplained hematuria at baseline study
- Systolic BP > 180 mmHg or diastolic BP > 100 mmHg at baseline study
- Systolic BP < 95 mmHg at baseline study
- Previous treatment with AF ablation
- Acute cardiovascular event [e.g., stroke, acute coronary syndrome (ACS), revascularization, decompensated HF, sustained ventricular tachycardia, return of spontaneous circulation (ROSC)] <8 weeks prior to baseline study
- Severe valvular disease or have prosthetic valve
- Treatment with chronic oral steroid (>30 consecutive days) at a dose equivalent to oral prednisolone $\geq$ 10 mg/d, within the past 1 month
- History of any malignancy within 5 years
- Clinically profound hepatic dysfunction
- Clinically uncontrolled thyroid dysfunction
- Patients incapable of completing the trial because of any severe medical condition by clinical decision
- Patients with poor compliance (defined as 80–120%), except for reasonable situations judged by physician

DM, diabetes mellitus; MODY, maturity onset diabetes of the young; EPI, epidemiology collaboration; BP, blood pressure; AF, atrial fibrillation; HF, heart failure.

pressure; (3) blood sampling for CBC, BUN/Cr, electrolyte, coagulation profile, HbA1c, fasting glucose, AST/ALT, lipid battery, and NT-proBNP; (4) urine sampling for dip stick urinalysis with microscopy and measuring proteinuria (especially for albuminuria); and (5) TTE to measure the LA size and left ventricular ejection fraction (LVEF). Any other additive assessing procedure may be conducted and recorded during the clinical course of the patients.

**Table 5. Data-collection requirements.**

| Investigation | Baseline | 3M | 6M | 9M | 12M |
|---|---|---|---|---|---|
| Inclusion/Exclusion criteria | X | | | | |
| Medical History | X | X | X | X | X |
| Physical Examination | X | X | X | X | X |
| 12-lead ECG | X | X | X | X | X |
| 24-h Holter ECG* | X | X | X | X | X |
| Blood, Urine sample | X | | | | X |
| Transthoracic Echocardiography (TTE) | X | | | | X |
| NT-proBNP | X | | | | X |
| EHRA Score | X | X | X | X | X |
| EQ-5D, SF-12 | X | | | | X |
| Adverse Events History** | | X | X | X | X |

*Patients with an ablation (RFA or 4PVI) will be monitored every 3 months that point on for an additional year.

**To evaluate adverse events, any suitable diagnostic investigation can be performed.

Abbreviations. ECG, electrocardiogram; NT-proBNP, N-terminal pro-B-type natriuretic peptide; EHRA, European Heart Rhythm Association; EQ-5D, EuroQol-5 Dimension; SF-12, Short Form-12 Health Survey Questionnaire.

During patient monitoring, if AF recurs and the patient complains of its symptoms, RFA or cryoablation (4PVI) will be performed at the discretion of the physician in accordance with the 2020 ESC [30] 2019 AHA/ACC/HRS guidelines [20] for AF. For patients with ablation, 24-h Holter ECG monitoring every 3 months for an additional year will be performed from that point on. If severe adverse events occur, the patient will be excluded from the study after a committee meeting. The main concern of the review is the recurrence of AF, which is the primary outcome of this trial.

## Statistical plan, sample size, and power determination

In this study, 716 patients will be enrolled from 7 tertiary medical centers. The sample size was determined based on the primary outcome, i.e., the SGLT-2 inhibitors should show 40% reduction in AF after a year of treatment [32] and 50% of AAD users should step up to the ablation therapy [33,34]. Assuming 10% follow-up loss, the sample size of 716 patients (358 in each group) is expected to achieve 80% power and an alpha-level of 5% to detect the difference. The study is powered to demonstrate the superior additive effect of SGLT-2 inhibitors over other hypoglycemic agents using the Chi square method. AF recurrence rate and ratio of sinus rhythm on 24-h holter EKG will be analyzed using Chi square test, and left atrial size, NT-pro BNP and quality of life (AFEQT) score on 12-month final follow-up will be analyzed by student t-test. Disease-free survival and overall survival during the follow-up period will be calculated using Kaplan-Meier method. Mono-variable and multi-variable cox regression analysis will be used to calculate hazard ratio of SGLT-2 group.

To reduce the selection bias, all subjects will be randomized to allocate the hypoglycemic agents according to random number table. Missing data is planned to be controlled using multiple imputation.

## Study status and organization

Ewha Womans University Mokdong Hospital is responsible for this trial, and the Institutional Review Board (IRB) of Ewha Womans University Mokdong Hospital approved this research plan, including the written consent form (S2, S3 and S4 File). In addition to Ewha Womans University Mokdong Hospital, at least seven other tertiary hospitals in South Korea are now preparing for IRB approval of each hospital with the same research plan.

This study complied with the fundamental spirit of the clinical trial management standards of the Helsinki Declaration (revised 2013) and the International Conference on Harmonisation of Technical Requirements for Registration of Pharmaceuticals for Human Use–Guidelines for Good Clinical Practice (ICH-GCP). The study was initiated after approval of the IRB in each medical center, and after approval, centers recruited patients and the therapies were administered following the appropriate study protocol and clinical standards.

Patients who met the inclusion criteria and voluntarily agreed to enroll in the study provided written informed consent, which was reviewed and approved by the IRB of each participating hospital after sufficient explanation about the research process. Personal identifiers were kept confidential by the researcher, and the research data was recorded with initials and research subject identification information was coded.

All adverse events reported by the subjects during a 3-month follow-up were reported at meetings between researchers four times a year and evaluated by an independent committee. Interruption of participation was decided when the above adverse reaction was a considerable side effect of the SGLT-2 inhibitors (dapagliflozin and empagliflozin), as informed by the Ministry of Food and Drug Safety, or was a serious adverse event unrelated to causality.

## Discussion

Recently, SGLT-2 inhibitors have been shown to exhibit beneficial effects beyond glucose control. SGLT-2 inhibitors were originally developed as hypoglycemic agents. They target SGLT-2 protein, which is responsible for 90% reabsorption of filtered glucose in the proximal convoluted tubules of the kidneys [35], resulting in diuresis. Previous studies have shown that SGLT-2 inhibitors suppress sympathetic overactivity [36] and oxidative stress [37]. DAPA-HF and EMPEROR-Reduced trial demonstrated that SGLT-2 inhibitors can reduce cardiovascular mortality in patients with HF [14,15]. Considering that AF and CHF are mutual risk factors, we designed a clinical trial based on the rationale that SGLT-2 inhibitors might be able to improve the clinical outcome of AF with DM by establishing the rhythm control of AF.

We will compare the recurrence rate of AF of the two groups after performing AAD administration and hypoglycemic treatment for at least 3 months as a primary outcome and evaluating the degree of rhythm control of AF. Because enlisting patients who were previously diagnosed with DM and whose glucose levels are currently well controlled in the trial can give rise to grave ethical issues, we recruited patients newly diagnosed with AF or DM within 1 year. We decided not to choose any specific SGLT-2 inhibitor agents and their dosage right away; instead, we decided to recommend physicians to utilize the maximum tolerable doses.

Several systemic reviews and meta-analyses have described advantages of catheter ablation over AAD in maintaining sinus rhythm [38–42] and improving the quality of life [43]. The CABANA trial showed that catheter ablation was superior in reducing the recurrence rate of AF compared to the AAD-only therapy [44]. Assuming that catheter ablation will improve the clinical outcome and lower the AF recurrence rate compared to that achieved with the AAD-only treatment in patients with AF and DM, we will add the following procedures after the random assignment of SGLT-2 inhibitors to minimize the bias caused by study protocols: First, the patient takes the assigned medicine for at least 3 months. Second, if AF recurrence is detected and the patient manifests clinical symptoms at follow-up visits, the physician will make a clinical decision on catheter ablation. AF recurrence after ablation (RFA and 4PVI) within a year will be measured by using the established protocol as the primary outcome among patients who have underwent ablation. In this setting, however, comparing the AF burden between the case (SGLT-2 inhibitor) and control (non-SGLT-2 inhibitor) groups at 12 months of trial would disturb the study results. Thus, the recurrence of AF is set as the primary outcome in each step (AAD and ablation).

Previous studies, including DAPA-HF, EMPERER-Reduced, and DECLARE-TIMI 58, have focused on the efficacy of SGLT-2 inhibitors in reducing the risk of adverse cardiovascular events as the primary endpoint in patients with DM, HF, or AF. However, the effects of SGLT-2 inhibitors on arrhythmia, especially AF, were only described by subgroup post-hoc analyses or meta-analyses, so they usually focused on the AF incidence rate and could not analyze other specific properties of AF [16,32,45]. Thus, our trial was advantageous because it allowed the measurement of AF control prospectively in patients with AF and DM.

Very few studies have quantitatively analyzed the AF burden, represented as the percentage of AF time during the monitoring period, because continuous ambulatory monitoring is needed to calculate the exact AF burden [22]. There are some ongoing clinical trials such as DAPA-AF that attempt to quantitatively evaluate the AF burden using cardiac implantable electronic devices (CIED). On the contrary, we plan to measure the AF burden after randomization of SGLT-2 inhibitors, before and after ablation, by utilizing 24-h Holter ECG, which is an excellent choice in terms of accessibility. Furthermore, our study results and information about the population with AF and DM will provide valuable data for subsequent studies in the future.

The LA size and blood level of NT-proBNP are set as the secondary endpoints, as we speculate that SGLT-2 inhibitors will delay the pathophysiologic course of AF. Remodeling of LA by itself acts as an underlying substrate for AF. In addition, progression of AF further deteriorates the LA function and accelerates LA remodeling, thereby continuing the vicious cycle [46,47]. Degree of abnormality of the LA structure and function can be used to determine the positive association with the electrical burden of AF [47]. Therefore, we plan to evaluate the progress of LA remodeling by measuring the degree of LA enlargement, i.e., change in the LA size by utilizing TTE. In addition, considering that NT-proBNP is often increased in AF and shows positive correlation with the clinical severity of AF [27,28], NT-proBNP can be used for assessing the severity of AF. Finally, the quality of life of all patients will be precisely measured.

The trial has some limitations as well. First, it primarily focuses on the effect of SGLT-2 inhibitors in patients newly diagnosed with AF and DM within 1 year. As a result, long-term effects of SGLT-2 inhibitors cannot be analyzed in this trial. Nonetheless, the short-term effects of SGLT-2 inhibitors are meaningful because these inhibitors reduced the incidence of atrial arrhythmia within 1 year compared to that in the control group [32] and the diuretic effect of SGLT-2 inhibitors appeared to be most prominent in first three months of administration only [48]. Second, a certain SGLT-2 inhibitor product is not unified in this trial. Physicians can choose the specific product and the maximum tolerable dose based on their clinical decisions and settings. To control the bias, a subgroup analysis including stratification on products will be conducted as post-hoc analysis.

## Conclusions

Data on the effect of SGLT-2 inhibitors derived from the trials that solely target patients with AF and DM have limited applicability. The proposed trial will evaluate the efficacy and safety of SGLT-2 inhibitors on the AF rhythm control in patients with AF and DM. The results generated by this trial will provide an invaluable dataset on rhythm control in AF with DM for future studies and offer novel information to assist in clinical decisions.

## Supporting information

**S1 File. SPIRIT checklist.**
(DOC)

**S2 File. Study protocol which was approved by institutional review board of Ewha Womans University Mokdong Hospital (original Korean version).**
(PDF)

**S3 File. Study protocol which was approved by institutional review board of Ewha Womans University Mokdong Hospital (English-transcribed version).**
(DOCX)

**S4 File. Approval latter from institutional review board of Ewha Womans University Mokdong Hospital (original Korean version).**
(PDF)

**S5 File. Approval latter from institutional review board of Ewha Womans University Mokdong Hospital (English-transcribed version).**
(PDF)

## Author Contributions

**Conceptualization:** Tae-Hoon Kim, Hee Tae Yu, Jung Myung Lee, Jin-Kyu Park, Yong-Soo Baek, Dong Hyeok Kim, Jaemin Shim, Boyoung Joung, Moon-Hyoung Lee, Hui-Nam Pak, Junbeom Park.

**Funding acquisition:** Junbeom Park.

**Investigation:** Tae-Hoon Kim, Hee Tae Yu, Jung Myung Lee, Jin-Kyu Park, Yong-Soo Baek, Dong Hyeok Kim, Boyoung Joung, Moon-Hyoung Lee, Hui-Nam Pak.

**Methodology:** Tae-Hoon Kim, Hee Tae Yu, Jung Myung Lee, Jin-Kyu Park, Yong-Soo Baek, Dong Hyeok Kim, Jaemin Shim, Boyoung Joung, Moon-Hyoung Lee, Hui-Nam Pak.

**Project administration:** Bo Kyung Jeon, Jaemin Shim, Junbeom Park.

**Supervision:** Junbeom Park.

**Validation:** Tae-Hoon Kim, Hee Tae Yu, Jin-Kyu Park, Yong-Soo Baek, Dong Hyeok Kim, Jaemin Shim, Boyoung Joung, Moon-Hyoung Lee, Hui-Nam Pak.

**Visualization:** Jung Myung Lee.

**Writing – original draft:** Kyuhyun Lee, Soo Kyoung Lee, Juyeon Lee.

**Writing – review & editing:** Bo Kyung Jeon, Junbeom Park.

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
