## [Decision Letter · Decision Letter 0]

26 Sep 2022

PONE-D-22-07177Protocol of BEYOND trial: Clinical BEnefit of Sodium-glucose cotransporter-2 (SGLT-2) inhibitors in rhYthm cONtrol of atrial fibrillation in patients with Diabetes mellitusPLOS ONE

Dear Dr. Park,

Thank you for submitting your manuscript to PLOS ONE. After careful consideration, we feel that it has merit but does not fully meet PLOS ONE’s publication criteria as it currently stands. Therefore, we invite you to submit a revised version of the manuscript that addresses the points raised during the review process. Please, carefully read and address all comments raised by the external reviewers. Please submit your revised manuscript by Nov 10 2022 11:59PM. If you will need more time than this to complete your revisions, please reply to this message or contact the journal office at plosone@plos.org. Please include the following items when submitting your revised manuscript:A rebuttal letter that responds to each point raised by the academic editor and reviewer(s). You should upload this letter as a separate file labeled 'Response to Reviewers'.A marked-up copy of your manuscript that highlights changes made to the original version. You should upload this as a separate file labeled 'Revised Manuscript with Track Changes'.An unmarked version of your revised paper without tracked changes. You should upload this as a separate file labeled 'Manuscript'.

We look forward to receiving your revised manuscript.

Kind regards,

Salvatore De Rosa

Academic Editor

PLOS ONE

“This trial is supported by Basic Science Research Program through the National Research Foundation of Korea (NRF) funded by the Ministry of Science, ICT & Future Planning (NRF-2017R1E1A1A01078382), and by the Korea Medical Device Development Fund grant funded by the Republic of Korea government (the Ministry of Science and ICT, the Ministry of Trade, Industry and Energy, the Ministry of Health & Welfare, the Ministry of Food and Drug Safety) (Project Number: 9991006899).”

“This trial is supported by Basic Science Research Program through the National Research Foundation of Korea (NRF) funded by the Ministry of Science, ICT & Future Planning (NRF-2017R1E1A1A01078382), and by the Korea Medical Device Development Fund grant funded by the Republic of Korea government (the Ministry of Science and ICT, the Ministry of Trade, Industry and Energy, the Ministry of Health & Welfare, the Ministry of Food and Drug Safety) (Project Number: 9991006899).”

4. We note that the original protocol that you have uploaded as a Supporting Information file contains an institutional logo. As this logo is likely copyrighted, we ask that you please remove it from this file and upload an updated version upon resubmission.

Please address all comments by external reviewers.

Reviewers' comments:

Reviewer's Responses to Questions

**Comments to the Author**

1. Does the manuscript provide a valid rationale for the proposed study, with clearly identified and justified research questions?

Reviewer #1: Yes

Reviewer #2: No

2. Is the protocol technically sound and planned in a manner that will lead to a meaningful outcome and allow testing the stated hypotheses?

Reviewer #1: Yes

Reviewer #2: Partly

3. Is the methodology feasible and described in sufficient detail to allow the work to be replicable?

Reviewer #1: Yes

Reviewer #2: Yes

4. Have the authors described where all data underlying the findings will be made available when the study is complete?

Reviewer #1: Yes

Reviewer #2: Yes

5. Is the manuscript presented in an intelligible fashion and written in standard English?

Reviewer #1: Yes

Reviewer #2: Yes

6. Review Comments to the Author

You may also provide optional suggestions and comments to authors that they might find helpful in planning their study.

Reviewer #1: The authors have clearly mentioned the details of ongoing trial protocols taking into account every aspect of the study and I have no major comments to make.

Just a minor comment:

1. The introduction section can be improvised a bit with a better flow of writing in the light of some recent findings albeit indirect but important.

"There is emerging evidence that suggests SGLT2I use may be associated with lesser episodes of AF compared to non-users. Example - incidental lesser AF amongst SGLT2I users as from USFDA AERS (Bonora et al. SGLT-2 inhibitors and atrial fibrillation in the Food and Drug Administration adverse event reporting system. Cardiovasc Diabetol 2021. https://doi.org/10.1186/s12933-021-01243-4) and, a pooled meta-analysis of 8 CV/Renal outcome trials (Okunrintemi et al. Sodium-glucose co-transporter-2 inhibitors and atrial fibrillation in the cardiovascular and renal outcome trials. Diabetes Obes Metab. 2021;23:276-280). These pieces of evidence suggest that an RCT is necessary to come to such a conclusion and a direct trial such as the BEYOND trial would help to this end.

Reviewer #2: The manuscript is inspiring and well-written, elaborating a clinical trial rationale in which authors tried to test out if SGLT-2 inhibitors might be able to improve the clinical outcome of AF with DM. However, there are some questions confusing me:

1. As mentioned in the context, AF and CHF are mutual risk factors, how to distinguish between AF patients and CHF patients (or patients with multiple risk factors for CHF)? I mean, CHF patients will more likely to benefit from SGLT-2I as shown in EMPEROR-Reduced and EMPEOR-Preserved, so, naturally, patients with CHF and AF are more likely to benefit from the SGLT-2 inhibitor, however, I did not see NT-proBNP or CHF in the exclusion criteria, is it reasonable to propose that a substantial number of patients in your inclusion process are complicated with CHF or multiple risk factors of CHF?

2. I am just wondering about whether AF evaluation will be presented in your following articles, such as AF type, symptom severity, severity of AF burden, stroke risk, etc. Since they are important baseline characteristics and should be balanced in the intervention and control groups.

7. PLOS authors have the option to publish the peer review history of their article (what does this mean?). If published, this will include your full peer review and any attached files.

Reviewer #1: **Yes: **Awadhesh Kumar Singh; MD; DM

Reviewer #2: No

---

## [Author Response · Author response to Decision Letter 0]

11 Nov 2022

- Respond to Journal Requirements -

A. We have re-checked and edited our manuscript set, including the naming of files to meet the PLOS ONE's style requirements.

“This trial is supported by Basic Science Research Program through the National Research Foundation of Korea (NRF) funded by the Ministry of Science, ICT & Future Planning (NRF-2017R1E1A1A01078382), and by the Korea Medical Device Development Fund grant funded by the Republic of Korea government (the Ministry of Science and ICT, the Ministry of Trade, Industry and Energy, the Ministry of Health & Welfare, the Ministry of Food and Drug Safety) (Project Number: 9991006899).”

“This trial is supported by Basic Science Research Program through the National Research Foundation of Korea (NRF) funded by the Ministry of Science, ICT & Future Planning (NRF-2017R1E1A1A01078382), and by the Korea Medical Device Development Fund grant funded by the Republic of Korea government (the Ministry of Science and ICT, the Ministry of Trade, Industry and Energy, the Ministry of Health & Welfare, the Ministry of Food and Drug Safety) (Project Number: 9991006899).”

A. We deleted all the funding-related text in the manuscript – in the Materials and Methods and Acknowledgements – as requested. We would like to retain the original Funding Statement.

A. We re-organized and reinforced the ethics statement that you requested in the ‘Study status and organization’ subsection of the Methods section. The exact name of the ethics committee who approved our study design – the Institutional Review Board of Ewha Womans University Mokdong Hospital – has been indicated. A statement regarding receiving informed written consent from patients has also been included.

4. We note that the original protocol that you have uploaded as a Supporting Information file contains an institutional logo. As this logo is likely copyrighted, we ask that you please remove it from this file and upload an updated version upon resubmission.

A. We have deleted the institutional logo in supplement files 2 and 3 and re-uploaded the modified files.

- Respond to Reviewers' comments -

Reviewer #1: The authors have clearly mentioned the details of ongoing trial protocols taking into account every aspect of the study and I have no major comments to make.

Just a minor comment:

1. The introduction section can be improvised a bit with a better flow of writing in the light of some recent findings albeit indirect but important.

"There is emerging evidence that suggests SGLT2I use may be associated with lesser episodes of AF compared to non-users. Example - incidental lesser AF amongst SGLT2I users as from USFDA AERS (Bonora et al. SGLT-2 inhibitors and atrial fibrillation in the Food and Drug Administration adverse event reporting system. Cardiovasc Diabetol 2021. https://doi.org/10.1186/s12933-021-01243-4) and, a pooled meta-analysis of 8 CV/Renal outcome trials (Okunrintemi et al. Sodium-glucose co-transporter-2 inhibitors and atrial fibrillation in the cardiovascular and renal outcome trials. Diabetes Obes Metab. 2021;23:276-280). These pieces of evidence suggest that an RCT is necessary to come to such a conclusion and a direct trial such as the BEYOND trial would help to this end.

A. We have briefly mentioned the merits of our prospective study on the relationship between SGLT-2 inhibitors and AF in the Discussion section. However, there was a lack of references to recent studies; we therefore revised and added reports of several recent findings in the Introduction and Discussion sections according to your much-appreciated advice.

Reviewer #2: The manuscript is inspiring and well-written, elaborating a clinical trial rationale in which authors tried to test out if SGLT-2 inhibitors might be able to improve the clinical outcome of AF with DM. However, there are some questions confusing me:

1. As mentioned in the context, AF and CHF are mutual risk factors, how to distinguish between AF patients and CHF patients (or patients with multiple risk factors for CHF)? I mean, CHF patients will more likely to benefit from SGLT-2I as shown in EMPEROR-Reduced and EMPEOR-Preserved, so, naturally, patients with CHF and AF are more likely to benefit from the SGLT-2 inhibitor, however, I did not see NT-proBNP or CHF in the exclusion criteria, is it reasonable to propose that a substantial number of patients in your inclusion process are complicated with CHF or multiple risk factors of CHF?

A. We did not include NT-proBNP or CHF separately, according to the exclusion criteria. Therefore, it is believed that registered subjects will be mixed with a certain percentage of groups with and without HF. We anticipated that this population will be able to produce results that better reflect the real world, and as you mentioned, HF, a strong confounding factor, can be considered in future analyses, such as multivariable Cox regression analysis or subgroup analysis.

2. I am just wondering about whether AF evaluation will be presented in your following articles, such as AF type, symptom severity, severity of AF burden, stroke risk, etc. Since they are important baseline characteristics and should be balanced in the intervention and control groups.

A. The secondary outcomes of our study included AF burden, sinus rhythm maintenance, and EHRA symptom score. Therefore, the subjects participating in our study will be checked for 12-lead ECG, 24-h Holter ECG, and EHRA symptom scores at the beginning and end of follow-up. Through these clinical data, baseline data and changes in subjects' AF characteristics can be presented at the result-reporting stage.

---

## [Decision Letter · Decision Letter 1]

28 Dec 2022

Protocol of BEYOND trial: Clinical BEnefit of Sodium-glucose cotransporter-2 (SGLT-2) inhibitors in rhYthm cONtrol of atrial fibrillation in patients with Diabetes mellitus

PONE-D-22-07177R1

Dear Dr. Park,

We’re pleased to inform you that your manuscript has been judged scientifically suitable for publication and will be formally accepted for publication once it meets all outstanding technical requirements.

Kind regards,

Salvatore De Rosa

Academic Editor

PLOS ONE

Additional Editor Comments (optional):

The authors addressed all comments presented during revision.

Reviewers' comments:

Reviewer's Responses to Questions

**Comments to the Author**

1. Does the manuscript provide a valid rationale for the proposed study, with clearly identified and justified research questions?

Reviewer #1: Yes

2. Is the protocol technically sound and planned in a manner that will lead to a meaningful outcome and allow testing the stated hypotheses?

Reviewer #1: Yes

3. Is the methodology feasible and described in sufficient detail to allow the work to be replicable?

Reviewer #1: Yes

4. Have the authors described where all data underlying the findings will be made available when the study is complete?

Reviewer #1: Yes

5. Is the manuscript presented in an intelligible fashion and written in standard English?

Reviewer #1: Yes

6. Review Comments to the Author

You may also provide optional suggestions and comments to authors that they might find helpful in planning their study.

Reviewer #1: Authors have clarified about the study protocol in full detail and answered the reviewers' queries quite satisfactorily.

7. PLOS authors have the option to publish the peer review history of their article (what does this mean?). If published, this will include your full peer review and any attached files.

Reviewer #1: **Yes: **Awadhesh Kumar Singh

---

## [Editor Report · Acceptance letter]

6 Jan 2023

PONE-D-22-07177R1 

Protocol of BEYOND trial: Clinical BEnefit of Sodium-glucose cotransporter-2 (SGLT-2) inhibitors in rhYthm cONtrol of atrial fibrillation in patients with Diabetes mellitus 

Dear Dr. Park:

I'm pleased to inform you that your manuscript has been deemed suitable for publication in PLOS ONE. Congratulations! Your manuscript is now with our production department. 

Kind regards, 

on behalf of

Dr. Salvatore De Rosa 

Academic Editor

PLOS ONE